# Plastomes of *Bletilla* (Orchidaceae) and Phylogenetic Implications

**DOI:** 10.3390/ijms231710151

**Published:** 2022-09-05

**Authors:** Shiyun Han, Rongbin Wang, Xin Hong, Cuilian Wu, Sijia Zhang, Xianzhao Kan

**Affiliations:** 1Anhui Provincial Key Laboratory of the Conservation and Exploitation of Biological Resources, Wuhu 241000, China; 2Institute of Chinese Medicine Resources, Anhui College of Traditional Chinese Medicine, Wuhu 241002, China; 3Institute of Bioinformatics, College of Life Sciences, Anhui Normal University, Wuhu 241000, China; 4Anhui Provincial Engineering Laboratory of Wetland Ecosystem Protection and Restoration, School of Resources and Environmental Engineering, Anhui University, Hefei 230601, China

**Keywords:** orchidaceae, *Bletilla*, phylogeny, chloroplast tRNA, codon usage and aversion, highly polymorphic regions

## Abstract

The genus *Bletilla* is a small genus of only five species distributed across Asia, including *B. chartacea*, *B. foliosa*, *B. formosana*, *B. ochracea* and *B. striata,* which is of great medicinal importance. Furthermore, this genus is a member of the key tribe Arethuseae (Orchidaceae), harboring an extremely complicated taxonomic history. Recently, the monophyletic status of *Bletilla* has been challenged, and the phylogenetic relationships within this genus are still unclear. The plastome, which is rich in both sequence and structural variation, has emerged as a powerful tool for understanding plant evolution. Along with four new plastomes, this work is committed to exploring plastomic markers to elucidate the phylogeny of *Bletilla*. Our results reveal considerable plastomic differences between *B. sinensis* and the other three taxa in many aspects. Most importantly, the specific features of the IR junction patterns, novel pttRNA structures and codon aversion motifs can serve as useful molecular markers for *Bletilla* phylogeny. Moreover, based on maximum likelihood and Bayesian inference methods, our phylogenetic analyses based on two datasets of Arethuseae strongly imply that *Bletilla* is non-monophyletic. Accordingly, our findings from this study provide novel potential markers for species identification, and shed light on the evolution of *Bletilla* and Arethuseae.

## 1. Introduction

As one of the most species-rich families in vascular plants, Orchidaceae possesses approximately 28,000 species from 736 recognized genera [1,2]. Over the years, numerous studies have been performed on the phylogeny of Orchidaceae [3,4,5,6,7,8]. To date, within the family, the following five subfamilies are currently recognized: Apostasioideae, Vanilloideae, Cypripedioideae, Orchidoideae and Epidendroideae [9]. Among the five subfamilies, Epidendroideae is the largest and most diverse, containing approximately 14 tribes [4,10]. Within the subfamily Epidendroideae, Arethuseae is a key tribe, undergoing variable taxonomic revisions, and currently consists of two subtribes (Arethusinae and Coelogyninae) [8,10,11,12].

*Bletilla* (subtribe Coelogyninae), with great medicinal importance, is a small genus of only five species distributed in Asia, including *B. chartacea*, *B. foliosa*, *B. formosana*, *B. ochracea* and *B. striata* [13,14,15]. Interestingly, despite the small number of species, this genus has a complicated taxonomic history, due to morphological similarities with other genera, e.g., *Arethusa* and *Bletia* [15]. As shown in Table 1, the genus *Bletilla* was subsequently included within the subtribe Bletieae, Bletilleae [16], Bletillinae [17], Bletiinae [10,12] and Coelogyninae [8], respectively. Among them, the most widely convincing taxonomic system for *Bletilla* is that of van den Berg et al. [4,8,9,18]. In this system, Arethuseae only included two subtribes (Arethusinae and Coelogyninae), and *Bletilla* was nested within Coelogyninae instead of Bletiinae.

Nevertheless, although most studies have focused on the systematic status of *Bletilla*, only a few reports have involved its internal branching patterns. Based on nrDNA-ITS and plastid gene *matK*, Li et al. [19] observed that the genus *Bletilla* was monophyletic, with *B. ochracea* being a sister to the remaining species of this clade; Feng et al. [20] found that *B. formosana* was a sister to *B. striata* + *B. ochracea*. Moreover, the monophyletic status of *Bletilla* has been challenged by a recent study from Huang et al. [18]. In this work, with combined morphological and molecular data, the species *B. foliosa* (a synonym of *B. sinensis*) has been treated as a new monotypic genus *Mengzia*. The monophyletic status of *Bletilla* and the phylogenetic relationships within this genus are still unclear. Thus, more evidences and further investigations are needed to clarify these issues.

As we know, the plastid genome (plastome), which is rich in both sequence and structural variation, has emerged as a powerful tool for understanding plant evolution [21,22,23,24,25,26,27,28]. In particular, with the rapid development of high-throughput sequencing technologies, numerous studies have enriched plastomic resources, such as the genomic composition, structural variation, high diversity regions [29,30,31]. Additionally, there are still several important characteristics of plastome, which harbor phylogenetic implications, that need to be explored further. For instance, studying plastomic tRNA (pttRNA), which accumulated multiple mutation events, is one point for delving into the plastid evolution [32]. It is becoming widely recognized that tRNAs possess highly conserved clover leaf-like structures [33]. Interestingly, our recent study surprisingly found some novel structures in the pttRNAs of the Macaronesian species (Crassulaceae) [28]. We further identified these secondary structural variations as genus-specific markers. Accordingly, exploring the pttRNAs’ secondary structure in *Bletilla* greatly benefit a better understanding of the phylogeny of this genus.

In addition, codon usage bias (CUB), referring to the unbalanced utilization of synonymous codons in coding DNA, can be analyzed for getting insights into the evolutionary patterns of both taxa and genes [34,35,36,37,38]. By the statistical analysis of CUB, we can speculate which factor is mainly responsible for bias pattern, usually mutational bias or natural selection [39,40,41]. It should also be noted that the CUB pattern has been reported to be highly associated with gene expression level [42,43]. Based on that, a codon that has a distinct positive relationship between its frequency and gene expression is defined as the optimal codon [44,45]. Codon optimality was attributed as a major determinant of mRNA stability [46,47,48]. Additionally, codon aversion motifs (CAM), presented as the nonuse of codons in genes, has recently been found to be a novel marker for phylogeny studies [49,50]. Definitely, the phylogenetic implications obtained from the analyses of the codon usage and aversion will improve our understanding of the phylogeny of *Bletilla*.

In this work, aiming to explore the interspecific differences of the genus *Bletilla*, four new plastomes were reported. Through comprehensive analyses, we are committed to address (1) the compositional variations of plastomes among the members of *Bletilla*; (2) the differences in novel secondary structures of pttRNAs; and (3) the phylogenetic relationships within the genus *Bletilla*. Ultimately, our findings of this study will shed light on the evolution of *Bletilla* and Arethuseae.

## 2. Results

### 2.1. Plastomic Organizations and Structural Features

The size of four complete plastome sequences of *Bletilla* ranged from 156,942 (*B. sinensis*) to 160,168 bp (*B. ochracea*), with typical quadripartite structures, containing LSC regions (86,289–87,746 bp) and SSC regions (18,249–18,804 bp) separated by two IR regions (26,202–26,809 bp each) (Figure 1, Table 2). The four plastomes were extremely conserved in GC contents and gene numbers, sharing the highest GC content in IR (43.2%), followed by LSC (35.0–35.1%) and SSC (30.2–30.4%). All investigated plastomes possessed 135 genes, containing 86 protein-coding genes (PCG), 8 rRNAs, 38 tRNAs and 3 pseudogenes (Appendix A).

As shown in Figure 2, the four *Bletilla* species displayed similar structures at JSA (junction IRA/SSC) and JSB (junction IRB/SSC), where *ycf1* (functional copy) and *ndhF* genes spanned these two junctions, respectively. Additionally, a 55 bp overlap between *ycf1* (pseudo copy) and ndhF was found at JSB in all four plastomes. Most notably, upon further analysis, we found that *B. sinensis* had several unique features. For instance, compared with the rather high similarity of plastome size among the three other species of *Bletilla* (159,484–160,022 bp), *B. sinensis* featured a much smaller size (156,942 bp). Moreover, we also detected that *rpl22* of *B. sinensis* completely existed in LSC regions, whereas this gene in the other three species was all across LSC and IRb. To better understand the structural variation of IR-LSC junctions, we further investigated a total of 16 well-annotated plastomes from the tribe Arethuseae. Additionally, the results show that the same location of *rpl22* was found in *B. sinensis* and all species in node 1. Meanwhile, all species in node 2 and 3, together with *Arundina graminifolia*, shared the same expansion of IR in *rpl22* gene (Figure 2). Moreover, we also observed that each IR of Arethuseae had a *trnH-rps19* gene cluster near both JSA and JSB.

In addition to coding regions, the diversity might also occur in the non-coding regions of the plastomes. Hence, we further analyzed DNA polymorphism among the four plastomes of *Bletilla*. To detect the intrageneric divergence, the calculation was conducted in two groups: all four *Bletilla* taxa (group A) and three *Bletilla* species (excluding *B. sinensis*) (group B). The identified highly polymorphic regions (HPR) of each group displayed a high degree of variation (Figure 3). In group A, a total of 12 regions were identified with the nucleotide divergence (Pi) ranging from 0.03667 to 0.07028 and 44–353 mutation sites. In comparison, group B possessed 14 HPRs featuring Pi in 0.00556–0.01185 and different sites in 5–33 (Table 3). Most interestingly, only four regions were shared by the two groups, and the Pi values of group A were overall almost 7 times larger than those of group B. Complicated relationships within *Bletilla* could be inferred by all the above divergence, and the unique patterns of the IR junction and HPR might serve as specific markers for this genus.

### 2.2. High-Informative Patterns of pttRNAs’ Secondary Structures

The secondary structures of pttRNAs were identified and compared among four plastomes of *Bletilla*. In 38 tRNA genes, a total of 12 novel putative pttRNA structures were detected (Figure 4) and were clustered into five groups: (1) pttRNA with one novel loop located in the accept stem: tRNA^Arg^-ACG and tRNA^Thr^-UGU; (2) pttRNA with one novel loop located in the D arm: tRNA^Met^-CAU; (3) pttRNA with novel variations in the anticodon arm: tRNA^Ile^-GAU harbored an intron in all four *Bletilla* species and tRNA^Gln^-UUG in *B. sinensis* contained an expanded anticodon loop; (4) pttRNA with novel structures in the variable loop: tRNA^Tyr^-GUA (with a 3-bp stem), as well as tRNA^Leu^-CAA, tRNA^Leu^-UAA, tRNA^Leu^-UAG, tRNA^Ser^-GCU and tRNA^Ser^-GGA (with new loops); (5) pttRNA with new structures in multiple regions: tRNA^Ser^-UGA had one new loop in the acceptor arm, T arm and variable loop, respectively.

To explore potential new markers for DNA barcoding, we further compared these novel putative pttRNA structures within *Bletilla*. Interestingly, the species *B. sinensis* has unique structural features of pttRNA. For example, the expanded 9-nt anticodon loop of tRNA^Gln^-UUG was found to be characteristic for *B. sinensis* (Figure 4) compared with the ordinary 7-nt loop for the remaining three *Bletilla* species. Moreover, in the variable loop, three unique pttRNAs were also detected (Figure 5): (1) *B. sinensis* only had one new variable loop in tRNA^Leu^-UAA, while the other three species all possessed two new loops; (2) tRNA^Ser^-GCU contained anticodon-like sequences in the variable loops for all four *Bletilla* species, however, the sequence 5′-UUU-3′ is only for *B. sinensis*, while it is 5′-UUA-3′ for the other three; (3) the position nt 56 in tRNA^Ser^-UGA was occupied by ‘C’ for *B. sinensis* and ‘A’ for the other three.

Furthermore, we conducted comparative analyses at the intergeneric level in the tribe Arethuseae (involving 22 released plastomes in total). Notably, except for tRNA^Ser^-UGA, the structural features of tRNA^Gln^-UUG, tRNA^Ser^-GCU and tRNA^Leu^-UAA of *B. sinensis* differed from the other species of the tribe Arethuseae.

### 2.3. Patterns of Codon Usage and Aversion

To explicate the patterns of codon usage and aversion among four species of *Bletilla* and one closely related species *Arundina graminifolia*, we performed the evaluation of the effective number of codons (ENCs), relative synonymous codon usage (RSCU) value, optimal codons and codon aversion motifs (CAM). Only 53 CDSs with sizes of at least 300 bp were considered for further analyses.

To reflect varying levels of CUB, the lowest and highest 5% of ENC values were selected and compared among these five species. As presented in Table 4, we detected high similarities in *B. formosana*, *B. ochracea* and *B. striata*, which shared the exactly same pattern for both low (*rps18*, *rpl16* and *psbD*) and high (*clpP*, *ndhE* and *ycf3*) groups. In contrast, for *B. sinensis*, the low group included *rps8*, and the high group possessed *rps4* and *ndhJ*. It was worth noting that the *B. sinensis* was found to share some similarities with *A. graminifolia* on ENC values. For instance, the *rps8* (37.85) gene was also one of the lowest in *A. graminifolia* and *ndhJ* (53.81) was the third highest as well.

Considering the codon aversion can act as a new character system in phylogenetics, we also employed the analysis of CAM for these 53 CDSs of five plastomes (Appendix A). The genus *Bletilla* had a highly conservative codon aversion pattern excluding *B. sinensis*, harboring the same motifs for 27 CDSs. Among these 27 CDSs, five are shared by *B. sinensis* (*clpP*, *ndhB*, *ndhJ*, *rps7* and *ycf4*). Surprisingly, we detected many unique CAM for *B. sinensis* from a total of 31 CDSs, indicating the substantial interspecific difference in *Bletilla*. Furthermore, two CDSs (*accD* and *atpF*) were observed that were shared by *A. graminifolia* and *B. sinensis*. Most significantly, as shown in Figure 6, the aversion motifs identified in *ndhA* and *rps11* genes could distinguish five investigated species.

The optimal codons identified in the five plastomes were shown in Table 5. Four species possessed four optimal codons, respectively: *B. formosana*, *B. ochracea* and *B. striata* had the same pattern (GGU, UUG, UCC and CGU), and *A. graminifolia* featured by GGU, UUG, UCC and CGA. Instead, *B. sinensis* only had two optimal codons (GGU and CGA), showing a high degree of diversity. Notably, GGU was shared by all five species, and CGA was shared by *B. sinensis* and *A. graminifolia*.

### 2.4. Phylogenetic Implications of Plastomes within Bletilla

To further clarify the evolutionary relationships within the genus *Bletilla*, especially the taxonomic position of *B. sinensis*, we performed phylogenetic analyses. Along with four new plastomes generated in this study, our phylogeny totally covered 36 species from 9 genera of Arethuseae using the datasets of 79 PCGs. Based on the 69,937-bp concatenated sequence, similar tree topologies were obtained for both ML and BI algorithms.

As Figure 7 displayed, most Arethuseae species formed two main clades. Clade I comprised 6 genera, which could be further divided into two subclades. All *Pleione* species clustered in a strongly supported monophyletic group (subclade A) ([BS] = 100, [PP] = 1.0). As sister to subclade A, subclade B consisted of four genera (*Pholidota*, *Coelogyne*, *Panisea* and *Bulleyia* ([BS] = 100, [PP] = 1.0). Among the members of subclade B, the two genera (*Pholidota* and *Coelogyne*) were recovered to be paraphyletic. *Thuniopsis*, a monotypic genus, was found to be the basal sister branch of Clade I ([BS] = 100, [PP] = 1.0). Furthermore, *Bletilla* species (excluded *B. sinensis*) and *Thunia alba* formed a distinct clade (clade II) (90 in ML, 1.0 in BI). Our data also support a sister relationship between *Arundina graminifolia* and these two main clades (clade I and II).

In addition, we also constructed a wider taxonomic sampled cladogram based on six cpDNA loci, additionally comprising the data of three Arethusinae taxa (Appendix A). Notably, clade I showed a highly similar topology to the PCG trees. Additionally, its sister clade (clade II) was composed of four species, three of which (*Arethusa bulbosa*, *Eleorchis japonica* and *Calopogon tuberosus*) formed a well-supported subclade (100 in ML, 1.0 in BI), and *A. graminifolia* was weakly grouped with them.

Significantly, it was interesting to note that *B. sinensis* was not clustered with other *Bletilla* members, and was located at the basal position of Arethuseae ([BS] = 100, [PP] = 1.0) in all trees instead. Thus, our results suggest that the genus *Bletilla* is paraphyletic.

## 3. Discussion

With the aim of clarifying the interspecific relationships in the genus *Bletilla*, this study reported the plastomes sequences of four species. Comprehensive analyses were performed within *Bletilla* and the tribe Arethuseae, including basic genomic properties of plastids, the structural features of IR boundaries, the predicted structures of pttRNAs, the patterns of codon usage and aversion, as well as phylogeny. Thereby, this work provides abundant molecular evidence for resolving the taxonomic issues in *Bletilla*, and also sheds light on the evolution of Arethuseae.

Characterized by single-parent inheritance, conservative organization, and a relatively slow-evolving rate, the plastome is widely recognized as a super-barcode for plant species discrimination and phylogenetic analyses [51,52,53]. From our analyses, both similarities and differences were detected between *B. sinensis* and three other *Bletilla* species. All four plastomes harbored exactly the same number of genes and similar GC contents. With the same location of the *ycf1* gene, all four plastomes possessed a pseudo *ycf1*. Nevertheless, compared to other species, the gene content of *B. sinensis* at JLB was highly divergent. According to Downie and Palmer [52], any mutation occurring in the structure or content of plastome possibly implicated phylogeny. In our recent study on plastome evolution [28], the *rps19* genes in all investigated taxa of Crassulaceae were located at the JLB, and were extended by 105 or 110 bp in the IRb. However, all *rps19* genes from members of the tribe Arethuseae were present in IRb. Moreover, the *trnH-rps19* gene cluster in IRs observed in this study was also present in most monocots, implying that the duplication of this cluster was prior to the divergence of monocot lineages [54].

Nucleotide mutations are cluster-distributed and manifested as “hotspots” in plastomes [55]. Consistent with previous studies [56,57,58,59,60], we also observed that genes in the IR regions were slower to evolve than those in the SC regions. To our knowledge, few mutational hotspots were found in the IR regions. For instance, Henriquez et al. [31] investigated five plastomes of Monsteroideae, which exhibited no hotspots in the two repeat regions. The decreased rates of substitution might result from gene conversion in IR regions [58,61,62]. More interestingly, Li et al. [63] observed that genes translocated into the IR region of fern plastomes not only reduced substitution rates, but also increased the GC content. In addition, the substantial increase in the Pi values in group A (*Bletilla* included *B. sinensis*) compared to group B (*Bletilla* excluded *B. sinensis*) indicated the faster substitution rate in *B. sinensis*. This finding implies that the three species (*B. ochracea*, *B. formosana* and *B. striata*) were closely related to each other and distantly related to *B. sinensis*. Dong et al. [64] found that highly variable chloroplast markers were suitable for evolutionary studies on angiosperms at low taxonomic levels. In our recent research on the plastome evolution of *Aeonium* and *Monanthes* (Crassulaceae) [28], we strongly recommended that the hotspots (highly polymorphic regions) of plastomes might have important implications for phylogeny, and could be used for the DNA barcoding of plants. Therefore, the identified hotspots loci in this study obviously possessed higher informative divergence, which would act as more efficient markers for the barcoding and phylogeny of *Bletilla*.

As we know, the complete plastome has significant genomic resources for untangling phylogenetic issues [65,66,67,68]. Linking the mRNA and protein, pttRNA occupies a critical part of chloroplast [32,69,70]. Brennan and Sundaralingam [71] pointed out that tRNAs embodies two categories on the basis of a variable-region size: type I contains a small loop with 4 or 5 nt, and type II has a larger size with a stem (3–7 bp) as well as a loop (3–5 nt). In this work, among the 38 pttRNAs of *Bletilla*, seven with a variable arm were examined to belong to type II, including tRNA^Leu^-CAA, tRNA^Leu^-UAA, tRNA^Leu^-UAG, tRNA^Ser^-GCU, tRNA^Ser^-GGA, tRNA^Ser^-UGA and tRNA^Tyr^-GUA. It is noteworthy that tRNA^Tyr^ of type II was considered to be unique to prokaryotes in an early study [72]. However, through intensive and extensive sampling, Sun and Caetano-Anollés [73] found that all Tyr specific tRNAs in both Archaea and Eukaryotes were also classified as type II. The long variable arm was presumed to be an ancient structure and was lost in more derived tRNAs [73].

More notably, plastomic tRNA^Leu^-UAA was proved to have important phylogenetic implications. As shown in Figure 7, Arethuseae could be sorted into five distinct categories based on the variable loop of tRNA^Leu^-UAA. Type A was shared by Clade I, consisting of the genera *Bulleyia*, *Coelogyne* (except for *C. corymbosa*), *Panisea*, *Pholidota* (except for *P. yunnanensis*) and *Pleione*; type B was featured in the genus *Bletilla* (except for *B. sinensis*), *C. corymbosa* and *P. yunnanensis*; while types C, D and E were unique to *Thunia alba*, *Arundina graminifolia* and *B. sinensis*, respectively. Based on these analyses, we strongly suggest that the genus *Bletilla* is non-monophyletic. Our results clearly indicate that the secondary structures of pttRNAs are of highly informative value for phylogenetic analyses. Thus, more in-depth studies are needed to better understand the evolutionary significance of pttRNAs in plants.

The CUB pattern is affected by multiple factors, such as selection on translation [74], gene size [75] and composition [76], as well as mutation and selection pressure [41,77]. The ubiquity of CUB in different genes or taxa makes it an ideal resource for investigating molecular evolution, gene expression, nucleotide composition, etc. [78,79]. As a vital index of CUB, ENC is widely associated with the gene expression level. The lower value of ENC means the stronger impact from CUB, and vice versa [80]. Additionally, it has been widely acknowledged that a higher level of expression is generally associated with a more powerful bias [81,82]. Notably, in comparison with three other species of *Bletilla*, our analysis revealed that *B. sinensis* varied considerably in ENC pattern, which might imply that the gene expression mode in *B. sinensis* is different. Another important indicator for gene expression is optimal codons. In fact, codon optimality has been proven to be a key determinant of mRNA stability [46,47,48]. Previous studies proposed that the number of optimal codons might be correlated with a different selection mode [81,83]. Generally, a positive selection will result in an increased number of preferred codons, while negative selection might cause a decrease. Significantly, we found that *B. sinensis* harbored the least number (two) of optimal codons among the five taxa investigated, suggesting that this species might undergo more pressure from purifying selection than others. On the other hand, codon aversion has recently been proposed to have extraordinarily potential value in phylogeny [28,49,50,84,85]. Remarkably, this study identified extremely divergent CAM in *B. sinensis* compared with other analyzed species. Moreover, following the method of Miller et al. [50], we further recovered the phylogeny of Arethuseae using CAM, exhibiting the same topology with the tree based on the PCGs of plastomes. Accordingly, our analyses on the patterns of codon usage and aversion confirmed the non-monophyly of *Bletilla* within the tribe Arethuseae.

In order to overcome the limitations of a few loci and the taxon sampling estimated, we employed two datasets (79 plastomic PCGs for 38 taxa, and 6 cpDNA regions for 49 taxa) to construct the cladogram of Arethuseae, respectively. Significantly, the non-monophyly of *Bletilla* was strongly supported by two different phylogenetic methods. We also found a heterogeneity in the relationship between *B. sinensis* and *Arundina graminifolia* compared to the work of Huang et al. [18]. The latter study found a sister relationship of them, while *B. sinensis* was located at the basal position of Arethuseae in the present study. Currently, there are a limited number of phylogenetic informative sites for the phylogeny of Arethuseae. Hence, to better evaluate the taxonomic status of *B. sinensis*, more samples are needed.

## 4. Materials and Methods

### 4.1. Sample Material, DNA Extraction, Sequencing and Annotations

The fresh leaves of four *Bletilla* species (*B. formosana*, *B. ochracea*, *B. sinensis* and *B. striata*) were gathered, and their specific locations are listed in Appendix A. The extraction of whole-genomic DNA was achieved using the Plant Genomic DNA kit (Tiangen, Beijing, China) according to CTAB method [86]. TruSeq DNA PCR-Free Library Prep Kit (Illumina, San Diego, CA, USA) was employed for library construction. Additionally, the resulting libraries were then sequenced through Illumina Novaseq 6000 with 150 paired-ends and 350 bp insert size.

The sequenced reads were quality assessed by FastQC and trimmed using Fastp v.0.11.0 [87,88]. The obtained reads were then assembled by GetOrganelle v.1.7.5.0 [89], taking the plastome of *Bletilla striata* (MT193723) as reference [90]. Gene annotation was conducted by GeSeq [91], and the annotation results were manually confirmed, with the BLAST program for coding sequences (CDSs) and tRNAscan-SE v.2.0.3 for pttRNA genes [92,93]. Lastly, the plastomes were visualized by Chloroplot [94].

### 4.2. Comparative Structural Analyses among the Plastomes of Bletilla

Comprehensive structural analyses were conducted comparatively among four members of the genus *Bletilla*. Firstly, the nucleotide composition of the plastomes was identified using Bioedit [95]. The boundaries at the junctions of IR and SC regions were checked and plotted manually. Additionally, the secondary structure of pttRNAs was then predicted by tRNAscan-SE v.2.0.3 [46].

### 4.3. Plastomic Codon Usage and Aversion Indices

To investigate the plastomic codon usage, CodonW v.1.4.2 was employed for the calculation of the value of ENC and RSCU [96]. The RSCU value was applied to quantify the degree of even use for each synonymous codon, with a value larger than 1 favoring the use of a codon and vice versa [97]. In the range of 20–61, ENC values usually denote the bias of codon usage, and the strong bias features with a low value [98].

Furthermore, to explore the deep correlation between CUB and gene expression, we conducted the ΔRSCU method to determine optimal codons in *Bletilla* [44,45]. Taking the ENC values of the CDSs as a substitute for expression degree, the highest and lowest 5% were categorized as the low and high group, respectively [40,81] Then, the optimal codons were sifted out with the ΔRSCU > 0.08 as well as the RCU (relative codon usage) value > 1 in the high group and <1 in the low group.

Moreover, to gain more informative genetic evidence, the codon aversion motifs, possessing strong phylogenetic implications, were extracted and manually checked using the CAM algorithm [50].

### 4.4. Phylogenetic Inferences

To untangle the controversy of the phylogeny of *Bletilla*, we performed phylogenetic analyses within the tribe Arethuseae. Additionally, two species of *Liparis*, from the closely related tribe Malaxideae of Arethuseae, served as outgroups. Additionally, two phylogenetic sampling strategies were employed in this study.

The first one was the inclusion of all available complete plastomic sequences of the tribe Arethuseae from NCBI, along with four new data from this study. The 79 plastomic CDSs from a total of 38 taxa formed the first dataset, which represented nine genera of the tribe Arethuseae (Appendix A).

Furthermore, six cpDNA regions (*ccsA*, *matK*, *psaB*, *rbcL*, *rpoC1* and *ycf1*) were selected for the second dataset. It consisted of 49 species, including eight additional species of Coelogyninae and three Arethusinae taxa compared to the first dataset (Appendix A).

The two datasets were aligned, respectively, with MAFFT, under the default settings [99]. SequenceMatrix was then used for the concatenation [100], with gaps as missing data. After that, two approaches were chosen for phylogenetic analysis: maximum likelihood (ML) and Bayesian inference (BI).

The ML trees were inferred by RAxML 8.2.12 [101]. Fifty runs and one thousand bootstrap replicates were executed under the models identified by PartitionFinder v.2.1.1 with the “-raxml” command line [102]. We also checked the convergence of each node by the “-I autoMRE” option. For BI analysis, the best models for the dataset were determined by ModelTest-NG [103]. Two simultaneous runs with four Markov chains each were run for 10 million generations (sampling every 100 generations), and Tracer 1.7.1 was used to assess the convergence [104].

## 5. Conclusions

In the context of the controversial intrageneric relationships within *Bletilla*, this study newly sequenced plastomes from four species of *Bletilla*, and performed comparative analyses among them. Interestingly, our results reveal considerable plastomic differences between *B. sinensis* and the other three taxa in many aspects. Most importantly, the specific features of the IR junction patterns, novel pttRNA structures and codon aversion motifs can serve as useful molecular markers for *Bletilla*. Furthermore, at the tribe level, the variable region of plastomic tRNA^Leu^-UAA and IR boundaries showed important phylogenetic implications for Arethuseae. Additionally, our phylogenetic analyses based on the two datasets, covering 36 species and 49 taxa of Arethuseae, respectively, suggested the non-monophyly of *Bletilla* with strong support. The convincing molecular evidence reported herein will provide novel potential markers for species identification, and achieve a more profound understanding for the evolution of *Bletilla* and Arethuseae.

## Figures and Tables

**Figure 1 ijms-23-10151-f001:**
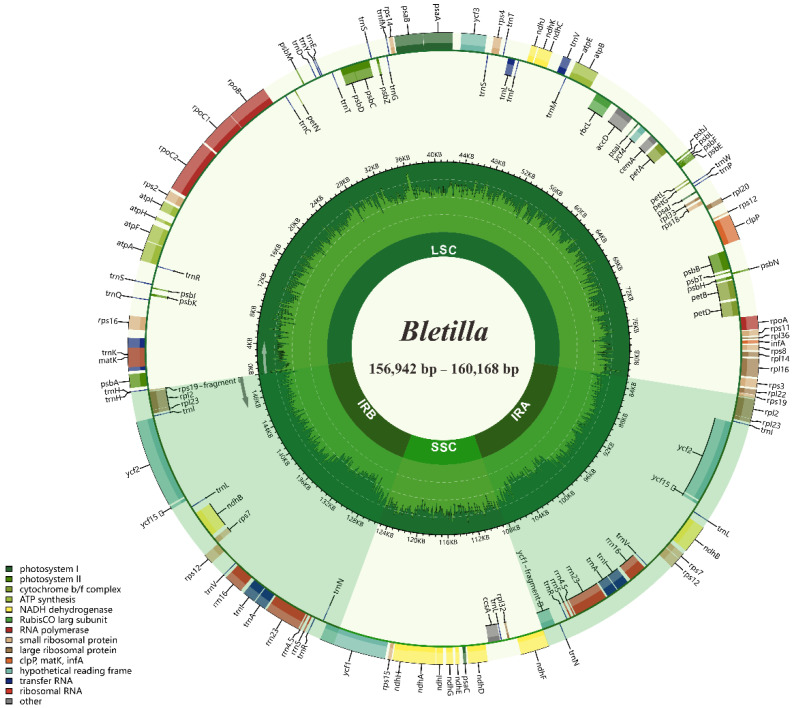
Chloroplast genome map of 4 *Bletilla* species. Directed with arrows, genes that are listed inside and outside of the circle are transcribed clockwise and counterclockwise, respectively. Genes are color-coded by their functional classification, with pseudogenes marked with asterisks. The GC content of the genome is depicted as the proportion of the shaded parts of each section.

**Figure 2 ijms-23-10151-f002:**
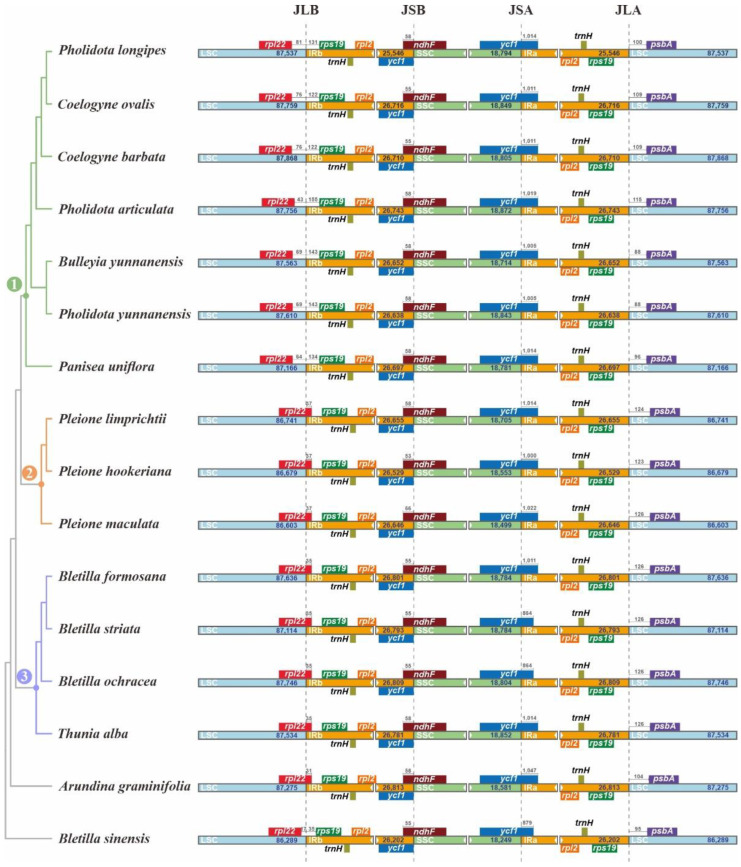
Contraction and expansion comparison of the junctions among 4 *Bletilla* plastomes. Blue, orange and green blocks represent the LSC, IR and SSC regions, respectively. Gene boxes represented above the block were transcribed clockwise and those represented below the block were transcribed clockwise. The base pairs (bp) number labeled within scale bars above the gene boxes indicate the extent of the integration between the junction region. The number 1, 2, and 3 above the branches represented different nodes, respectively.

**Figure 3 ijms-23-10151-f003:**
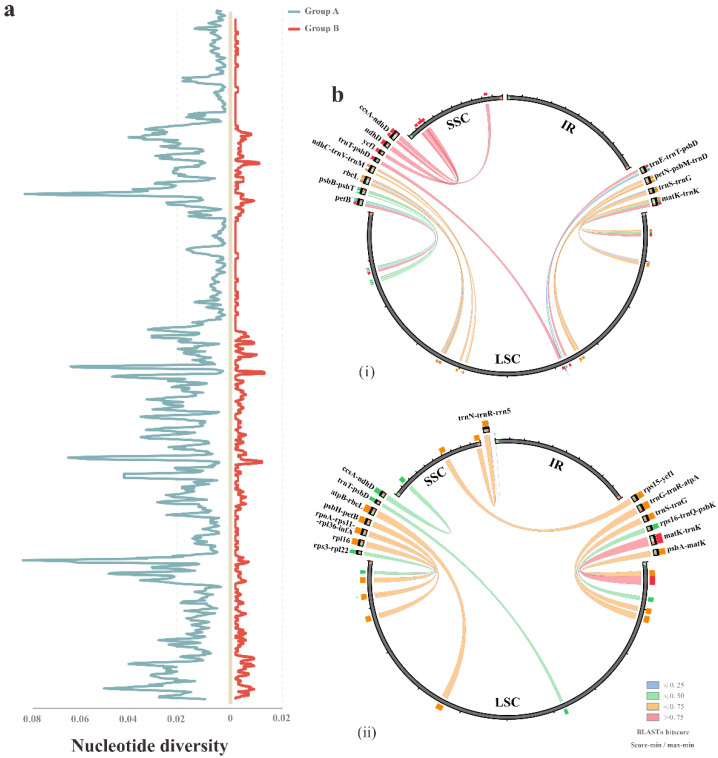
The HPR analyses among the plastomes of different groups. (**a**) The nucleotide diversity of the complete chloroplast genomes based on the comparison of group A (shown left the *Y axis*) and group B (shown right the *Y axis*), respectively. (**b**) The HPRs possessed by the two groups were shown in the Circoletto plot. The width of line reflected the region size, and the color is based on the BLASTn score: (**i**) group A; (**ii**) group B.

**Figure 4 ijms-23-10151-f004:**
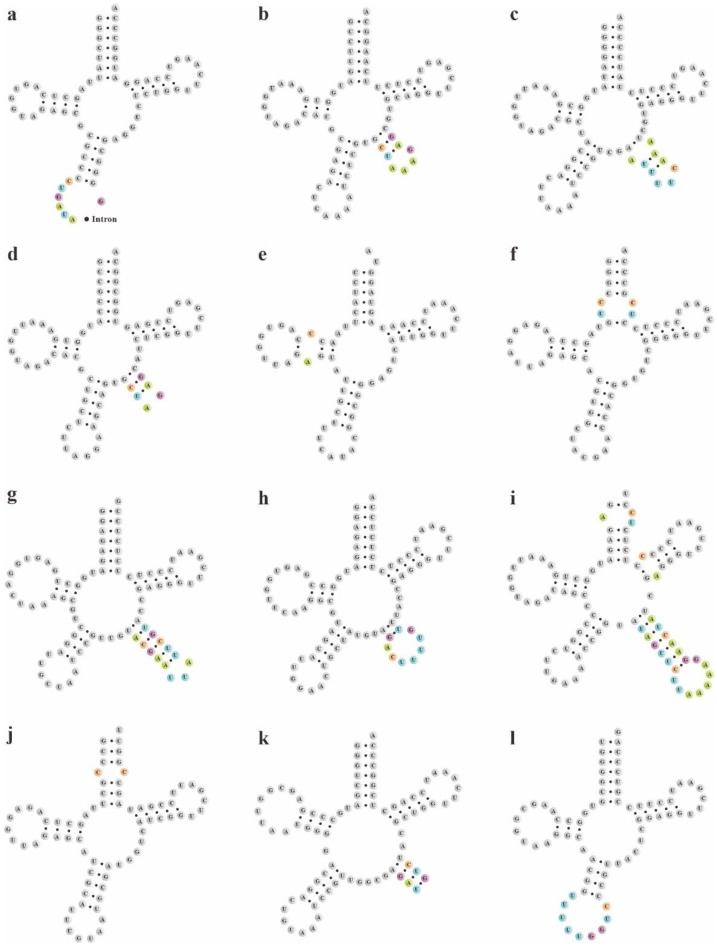
The novel putative pttRNAs structures in 4 *Bletilla* species. Specific locations of new structures were highlighted with color-coded nucleotides: (**a**) tRNA^Ile^-GAU; (**b**) tRNA^Leu^-CAA; (**c**) tRNA^Leu^-UAA; (**d**) tRNA^Leu^-UAG; (**e**) tRNA^Met^-CAU; (**f**) tRNA^Arg^-ACG; (**g**) tRNA^Ser^-GCU; (**h**) tRNA^Ser^-GGA; (**i**) tRNA^Ser^-UGA; (**j**) tRNA^Thr^-UGU; (**k**) tRNA^Tyr^-GUA; and (**l**) tRNA^Gln^-UUG (only in *B. sinensis*).

**Figure 5 ijms-23-10151-f005:**
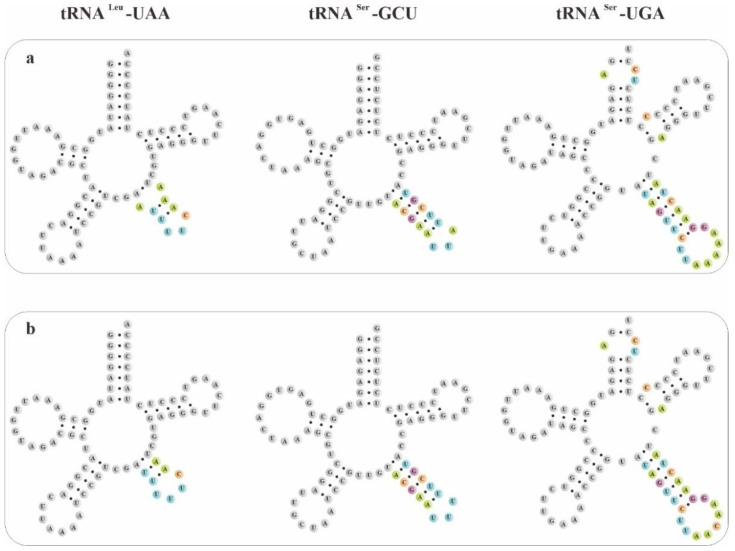
Three types of pttRNA were found to be different between *B. sinensis* and the other three *Bletilla* species: (**a**) *B. formosana*, *B. ochracea* and *B. striata*; and (**b**) *B. sinensis*.

**Figure 6 ijms-23-10151-f006:**
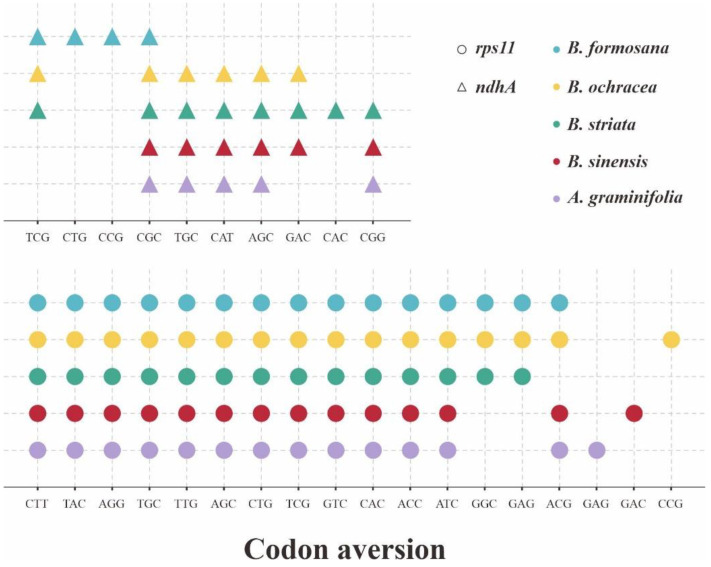
The species-specific codon aversion motifs of *ndhA* and *rps11* gene for the 5 investigated plastomes. The dots that were marked, respectively, in different colors, represent specific species.

**Figure 7 ijms-23-10151-f007:**
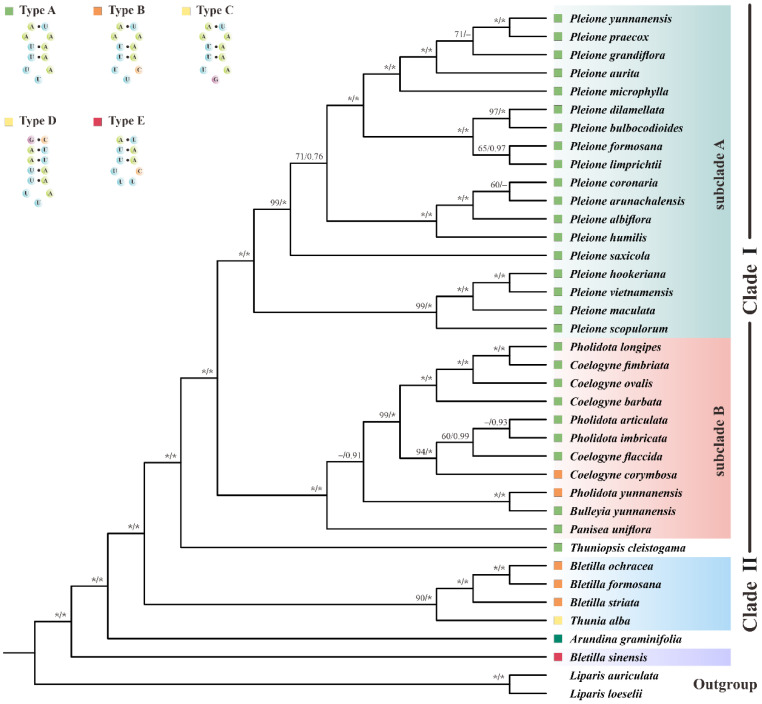
The phylogenetic tree of 36 Arethuseae species based on 79 plastomic PCGs using the maximum likelihood (ML) and Bayesian inference (BI) method. The bootstrap (BS) and Bayesian posterior probability (PP) values of each node were labeled (* denoted 100% bootstrap or 1.00 PP, with the omission of those <50% bootstrap or <0.5 PP).

**Table 1 ijms-23-10151-t001:** The taxonomic history of *Bletilla*.

Botanists	Taxonomic Status of *Bletilla*	
Bentham (1883)	Included within the genus *Bletia* Position: tribe Epidendreae, subtribe Bletieae	
Schlechter (1926)	As an individual genus associated with the genus *Arethusa* in the same subtribe (Bletia was placed in the subtribe Phajeae of tribe Kerosphaereae) Position: tribe Polychondreae, subtribe Bletilleae	
Mansfeld	*Bletilla*, *Arethusa* and *Bletia* were classified in the same tribe	
(1937, 1954)	Position: tribe Epidendreae	subtribe Bletillinae (including *Bletilla* and *Arethusa*)
		subtribe Phaiinae (including *Bletia*)
Dressler	*Bletilla*, *Arethusa* and *Bletia* were classified in the same tribe	
(1981, 1993)	Position: tribe Arethuseae	subtribe Arethusinae (*Arethusa*)
		subtribe Bletiinae (included *Bletilla* and *Bletia*)
Szlachetko (1995)	*Bletilla* and *Bletia* were classified in the same tribe (*Arethusa* was still placed in tribe Arethuseae) Position: tribe Bletiinae (including *Bletilla* and *Bletia*)	
Berg (2005)	*Bletilla* and *Arethusa* were classified in the same tribe (*Bletia* was placed in the subtribe Bletiinae of tribe Epidendreae)	
Until now	Position: tribe Arethuseae	subtribe Arethusinae (including *Arethusa*)
		subtribe Coelogyninae (including *Bletilla*)

**Table 2 ijms-23-10151-t002:** Comparative analysis among the plastomic features of the species in this study.

Species (Accession Number)		*B. formosana* (OP104328)	*B. ochracea* (OP104329)	*B. striata* (OP104330)	*B. sinensis* (MT806143)
Size (base pair, bp)	Total	160,022	160,168	159,484	156,942
	LSC	87,636	87,746	87,114	86,289
	IR	26,801	26,809	26,793	26,202
	SSC	18,784	18,804	18,784	18,249
Number of genes	Total	135	135	135	135
	PCGs	86	86	86	86
	tRNA	38	38	38	38
	rRNA	8	8	8	8
	Pseudo	3	3	3	3
	Intron-containing	17	17	17	17
GC content (%)	Total	37.2	37.2	37.2	37.2
	LSC	35.1	35.0	35.0	35.0
	IR	43.2	43.2	43.2	43.2
	SSC	30.3	30.3	30.2	30.4
	All genes	39.7	39.8	39.7	39.7
	CDS	38.0	38.0	38.0	37.9
	tRNA	53.0	53.1	53.2	53.2
	rRNA	54.9	54.9	54.9	54.8

**Table 3 ijms-23-10151-t003:** The highly polymorphic regions identified in the plastomes of the two groups.

Taxa		Region	Nucleotide Diversity	No. of Mutations	Region Length
Group A	1	*matK-trnK-UUU*	0.04243	199	1307
	2	*trnS-GCU-trnG-GCC*	0.03875	93	965
	3	*petN-psbM-trnD-GUC*	0.04389	158	1127
	4	*trnE-UUC-trnT-GGU-psbD*	0.07028	253	1331
	5	*trnT-GGU-psbD*	0.03667	44	731
	6	*ndhC-trnV-UAC-trnM-CAU-atpE*	0.04194	50	1540
	7	*rbcL*	0.05880	203	1247
	8	*psbB-psbT*	0.04569	109	1078
	9	*petB*	0.05546	188	1130
	10	*ccsA-ndhD*	0.05944	353	1504
	11	*ndhD*	0.03764	89	857
	12	*ycf1*	0.04000	48	621
Group B	1	*psbA-matK*	0.00556	15	1000
	2	*matK-trnK-UUU*	0.00612	33	1601
	3	*rps16-trnQ-UUG-psbK*	0.00556	10	845
	4	*trnS-GCU-trnG-GCC*	0.00667	18	1016
	5	*trnG-GCC-trnR-UCU-atpA*	0.00695	25	1205
	6	*trnT-GGU-psbD*	0.00556	5	609
	7	*atpB-rbcL*	0.00917	33	1257
	8	*psbH-petB*	0.01185	32	1009
	9	*rpoA-rps11-rpl36-infA*	0.00778	21	1000
	10	*rpl16*	0.00741	20	1023
	11	*rps3-rpl22*	0.00556	5	601
	12	*ccsA-ndhD*	0.00723	13	849
	13	*rps15-ycf1*	0.00926	25	1000
	14	*trnN-GUU-trnR-ACG-rrn5-rrn4.5-rrn23*	0.00630	17	1009

**Table 4 ijms-23-10151-t004:** The highest and lowest 5% ENC values of 53 CDSs among the 5 investigated species.

	*Bletilla formosana*		*Bletilla ochracea*		*Bletilla striata*		*Bletilla sinensis*		*Arundina graminifolia*	
	Gene	ENC	Gene	ENC	Gene	ENC	Gene	ENC	Gene	ENC
Low group	*rps18*	36.65	*rps18*	36.65	*rps18*	36.65	*rps18*	38.16	*rps8*	37.85
	*rpl16*	41.39	*rpl16*	41.39	*psbD*	42.05	*rps8*	40.80	*rps14*	42.03
	*psbD*	42.05	*psbD*	42.05	*rpl16*	42.20	*rpl16*	41.67	*petD*	42.03
High group	*clpP*	57.86	*clpP*	57.86	*clpP*	57.86	*ndhJ*	53.81	*ndhJ*	54.68
	*ndhE*	59.50	*ndhE*	59.50	*ndhE*	59.50	*rps4*	54.04	*clpP*	57.69
	*ycf3*	60.75	*ycf3*	60.75	*ycf3*	60.75	*clpP*	60.36	*ndhE*	57.83

**Table 5 ijms-23-10151-t005:** The optimal codons of the 5 investigated species.

Taxa	Optimal Codons				
*B. formosana*	GGU		UUG	UCC	CGU
*B. ochracea*	GGU		UUG	UCC	CGU
*B. striata*	GGU		UUG	UCC	CGU
*B. sinensis*	GGU	CGA			
*A. graminifolia*	GGU	CGA	UUG	UCC	

## Data Availability

The four plastomes’ sequences data generated in this study are available in GenBank of the National Center for Biotechnology Information (NCBI) (https://www.ncbi.nl-m.nih.gov/nuccore, accessed on 29 July 2022) under the access numbers: OP104328-OP104330, and MT806143.

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
