# Peer review of "Plastomes of Bletilla (Orchidaceae) and Phylogenetic Implications"

_ijms, 2022, doi:10.3390/ijms231710151_

Round 1

Reviewer 1 Report

Bletilla  is a small Asiatic genus with five species, including B. striata, B. ochracea, B. formosana, Bletilla foliosa and Bletilla chartacea, primarily distributed in Asia from southern and eastern China to Japan and Korea. This genus was described from a species originally placed in the American genus Bletia of the tribe Epidendreae. Subsequently, Bletilla has generally been associated with Arethusa in the tribe Neottieae. Some recent authors have transferred the Arethusa alliance from the Neottieae to the Epidendreae, once again placing Bletilla in the same tribe with Bletia, although in separate subtribes. Recently, the monophyletic status of Bletilla has been challenged, and the phylogenetic relationships within this genus are still unclear. This diversity of treatment reflects the complexity of the very large family Orchidaceae with its reticulate relationships within and between groups. In most classifications of the Orchidaceae, excessive importance has been attached to morphological features, including gynostemium details, that often prove fallible. As a result, closely related genera are separated, thereby obscuring evolutionary trends within the family. This appears to be the case with the genus Bletilla, which may be representative of a significant stage in the evolution of the groups in which Arethusa and Bletia have been classified. Anyway, the classification system of the genus Bletiila has not been clearly explained so far, so I consider the article submitted for review to be very important and necessary to explain the complex relationships inside the Orchidaceae.

In my opinion, this is an excellent article that organizes the Bletilla classification system. I consider the statement that the genus Bletilla is not monophyletic, based on the results of analyzes, to be a particularly valuable result. It is true that such hypotheses have already appeared in the past, but no one has proven them so far. The authors clearly showed that the secondary structures of pttRNAs show a high informative value for phylogenetic analyzes.

I recommend publishing this article in current version because it is methodologically sound and prepared perfectly. I would also like to congratulate the authors, it is a very well-designed and executed research project!

Round 2

Reviewer 2 Report

The authors have addressed all my previous concerns.